# Early Life Polychlorinated Biphenyl 126 Exposure Disrupts Gut Microbiota and Metabolic Homeostasis in Mice Fed with High-Fat Diet in Adulthood

**DOI:** 10.3390/metabo12100894

**Published:** 2022-09-23

**Authors:** Yuan Tian, Bipin Rimal, Wei Gui, Imhoi Koo, Philip B. Smith, Shigetoshi Yokoyama, Andrew D. Patterson

**Affiliations:** 1Department of Veterinary and Biomedical Sciences, The Pennsylvania State University, University Park, PA 16802, USA; 2Huck Institutes of the Life Sciences, The Pennsylvania State University, University Park, PA 16802, USA

**Keywords:** 3,3′,4,4′,5-pentacholorobiphenyl, early life exposure, high-fat diet, obesity, glucose homeostasis, metabolomics, gut microbiota

## Abstract

Evidence supports the potential influence of persistent organic pollutants (POPs) on the pathogenesis and progression of obesity and diabetes. Diet-toxicant interactions appear to be important in diet-induced obesity/diabetes; however, the factors influencing this interaction, especially the early life environmental exposure, are unclear. Herein, we investigated the metabolic effects following early life five-day exposure (24 μg/kg body weight per day) to 3,3′,4,4′,5-pentacholorobiphenyl (PCB 126) at four months after exposure in mice fed with control (CTRL) or high-fat diet (HFD). Activation of aryl hydrocarbon receptor (AHR) signaling as well as higher levels of liver nucleotides were observed at 4 months after PCB 126 exposure in mice, independent of diet status. Inflammatory responses including higher levels of serum cytokines and adipose inflammatory gene expression caused by early life PCB 126 were observed only in HFD-fed mice in adulthood. Notably, early life PCB 126 exposure worsened HFD-induced impaired glucose homeostasis characterized by glucose intolerance and elevated gluconeogenesis and tricarboxylic acid (TCA) cycle flux without worsening the effects of HFD related to adiposity in adulthood. Furthermore, early life PCB 126 exposure resulted in diet-dependent changes in bacterial community structure and function later in life, as indicated by metagenomic and metabolomic analyses. These data contribute to a more comprehensive understanding of the interactions between diet and early life environmental chemical exposure.

## 1. Introduction

Persistent organic pollutants (POPs) are toxic chemicals that are resistant to environmental degradation [1]. Most POPs are hydrophobic and poorly metabolized, thus making such compounds susceptible to bioaccumulation in animal and human fat [2]. POPs are toxic to human health, causing developmental defects, metabolic diseases, cancer, and, in some cases, death [3,4,5]. 

Concerns related to POP exposure were originally related to organochlorine pesticides, such as dichlorodiphenyltrichloroethane, but more recently, attention has been focused on POPs of industrial origin, most notably, polychlorinated biphenyls (PCBs) [6]. PCB 126 (3,3′,4,4′,5-pentacholorobiphenyl) is the most potent planar dioxin-like toxicant among the class of PCBs, which has a similar structure and toxicity to those of polychlorinated dibenzo-*p*-dioxins capable of activating the aryl hydrocarbon receptor (AHR) [7]. Unchanged PCB 126 may remain in the body and be stored for years, mainly in the fat and liver [8]. Accumulating evidence supports the hypothesis that PCB 126 exposure is associated with metabolic disruption [9,10,11]. A single dose of PCB 126 (326 μg/kg body weight) was shown to rapidly and significantly increase hepatic microsomal cytochrome P450 (CYP1A1) enzyme activity and perturb redox and metal homeostasis and antioxidant and enzyme levels, such as glutathione peroxidase activities in rat liver [12]. Long-term PCB 126 exposure induced systemic effects associated with diabetes, liver disease, and microbiota dysbiosis in rat and mouse models [9,10,11,13]. However, the long-term consequences of PCB 126 exposure at vulnerable life stages, such as early childhood, have received less attention [14]. Recently, we demonstrated that early life PCB 126 exposure resulted in metabolic abnormalities and microbiota changes in mice fed with standard chow diet in adulthood, providing evidence for an association between early life environmental pollutant exposure and increased risk of metabolic disorders later in life [15]. 

The increased risk of diabetes/obesity is positively correlated to PCBs exposure [16,17,18]. Studies have reported that 2,2′,4,4′,5,5′-hexachlorobiphenyl (PCB 153), a non-dioxin like PCB, is a diet-dependent obesogen, which may worsen non-alcoholic fatty liver disease via adipokine dysregulation and altered hepatic lipid metabolism in mice [19]. A recent study reported that PCB 126 exposure altered the global epi-transcriptome in a mouse model of non-alcoholic steatohepatitis [20]. Recent study also indicated that two doses of PCB 126 exposure (326 μg/kg body weight) dramatically disrupted the gut microbiota and host metabolism and increased intestinal and systemic inflammation in a low density lipoprotein receptor knockout (*Ldlr−/−*) mouse model fed with a high-cholesterol diet [21]. Likewise, diet-toxicant interactions appear to be important in diet-induced obesity/diabetes but remain largely unexplored, particularly for early life environmental exposure. 

In this study, mice at four weeks old were exposed to PCB 126 (24 μg/kg body weight per day) via the diet for five consecutive days, and we examined the PCB-induced changes at four months after exposure in mice eating a control (CTRL) or high-fat diet (HFD). Here, we chose a dose lower than those often utilized in PCB studies (300 μg/kg–50 mg/kg body weight) [12,19,21]. The purpose of this study was (*i*) to determine whether early life PCB 126 exposure worsens diet-induced metabolic diseases in adulthood by exacerbating previously implicated mechanisms, such as impaired glucose tolerance or perturbed lipid metabolism, and (*ii*) to determine the persistent and distinct changes of microbiota caused by early life PCB 126 exposure in two different diets.

## 2. Materials and Methods

### 2.1. Chemicals

Sodium 3-(trimethylsilyl) [2,2,3,3-^2^H_4_] propionate (TSP), D_2_O (99.9% in D), and PCB 126 were ordered from Cambridge Isotope Laboratories, Inc. (Tewksbury, MA, USA). Bile acid standards and deuterated internal standards were obtained from Sigma-Aldrich (St Louis, MO, USA) and Cayman Chemical (Ann Arbor, MI, USA). 

### 2.2. Animals and Diets

Three-week-old male C57BL/6J wild-type mice were obtained from Jackson Laboratories (Bar Harbor, MN, USA). High-fat diet (HFD) F3282 and control diet (CTRL) F4031 (Bio-Serv, Flemington, NJ, USA) were used to evaluate the effects of diet-toxicant interactions on mice (for more nutrient composition information about diets, see Appendix A). After acclimatization, mice were trained to eat transgenic bacon-flavored dough (Bio-Serve, Flemington, NJ, USA), formed into pills using a tablet mold, for one week. After training, mice at four weeks old were fed pills containing PCB 126 (a final dose of 24 µg/kg) or acetone (vehicle) continuously for five days (one pill per mouse per day). The mice (six mice per group) were sacrificed without fasting at four months after PCB 126 exposure (Figure 1A and Appendix A). The body weight was recorded every week, and urine and feces were collected before sacrifice. Blood, liver, cecal content, visceral white adipose, and intestinal tissue samples were collected immediately after sacrifice and kept at −80 °C. Animal experiments were performed using protocols approved by the Pennsylvania State University Institutional Animal Care and Use Committee (PROTO202001416).

### 2.3. Histopathology

Formalin-fixed liver tissues were embedded in paraffin wax, sectioned, and stained with hematoxylin and eosin (H&E). Oil red O staining of liver sections was performed on optimal cutting temperature (OCT)-embedded frozen liver sections by Histoserv, Inc. (Germantown, MD, USA). 

### 2.4. Blood Clinical Biochemistry and Cytokine Analysis

Liver injury markers, including serum alanine transaminase (ALT) and alkaline phosphatase (ALP), were assessed using the VetScan VS2 Chemistry Analyzer and the Mammalian Liver Profile (Abaxis Inc., Union City, CA, USA). Serum cytokine levels were assessed with a BioPlex 200 mouse cytokine array/chemokine array 32-Plex by Eve Technologies (Calgary, AB, Canada).

### 2.5. Glucose Tolerance Test

The glucose tolerance test was performed as previously described [22] and followed a 6 h fast. In brief, blood glucose was measured following intraperitoneal injection of 40% glucose (2.0 g/kg body weight). The blood glucose concentrations were measured using OneTouch Ultra 2 Meter (LifeScan, Malvern, PA, USA).

### 2.6. Liver Triglyceride and Glutathione Quantification

Liver triglyceride was measured using the Triglyceride Colorimetric Assay Kit according to the manufacturer’s instructions (Cayman Chemical, Ann Arbor, MI, USA). The ratios of reduced glutathione (GSH) and oxidized glutathione (GSSG) were measured with the GSH/GSSG Ratio Detection Assay Kit according to the manufacturer’s recommendations (Abcam, Cambridge, UK).

### 2.7. Tissue RNA Isolation and qPCR

Total RNA was isolated from liver, adipose, and intestine tissues by TRIzol reagent (Invitrogen, Carlsbad, CA, USA) according to the manufacturer’s protocol. cDNA was synthesized from 1 μg of RNA using qScript cDNA SuperMix (Quanta Biosciences, Gaithersburg, MD, USA). Quantitative PCR (qPCR) reactions were performed using SYBR green QPCR master mix with QuantStudio 3 Real-Time PCR system (Thermo Fisher Scientific, Waltham, MA, USA). The primers used were listed in Appendix A. The data were normalized to *Gapdh* mRNA levels using the ΔΔC_T_ method. 

### 2.8. ^1^H NMR Based Metabolomics Experiments

Sample preparations for cecal content, liver, and urine samples were performed as previously described [23]. All ^1^H NMR spectra were recorded using a Bruker Avance NEO 600 MHz spectrometer equipped with an inverse cryogenic probe (Bruker Biospin, Germany) at 298 K. NMR spectra were acquired with a typical 1D NMR spectrum named NOESYPR1D. The metabolites were confirmed with a set of 2D NMR spectra and published results [23,24]. ^1^H NMR spectra for liver lipids were corrected for phase and baseline distortions, referenced to TMS (*δ* = 0.0), and integrated using TopSpin 3.6 and AMIX 3.9 (Bruker Biospin, Ettlingen, Germany). The absolute quantification for lipid classes in liver were calculated as previously reported [24]. The spectra process and quantification for liver hydrophilic and cecal bacterial metabolites were performed using Chenomx NMR Suite (Chenomx Inc., Edmonton, AB, Canada). 

### 2.9. Bile Acid Quantitation by UPLC-MS/MS

Quantitative analysis of bile acids in feces was performed with an Acquity UPLC system coupled to a Waters Xevo TQS MS with an ACQUITY C8 BEH (2.1 × 100 mm, particle size 1.7 µm) UPLC column (all from Waters, Milford, MA, USA) [25]. The analytes were detected by multiple reaction monitoring (MRM) or selected ion monitoring (SIM) and normalized by their respective internal deuterated standard. The results were quantified by comparing the integrated peaks against a standard curve. 

### 2.10. Metagenomic Analysis

DNA was extracted from cecal contents using the E.Z.N.A. Stool DNA Kit (Omega Bio-Tek Inc., Norcross, GA, USA). Cecal DNA samples were submitted to the Pennsylvania State University Genomics Core Facility for NextSeq Mid-Output 150 × 150 paired end sequencing. The obtained demultiplexed reads were checked for quality using FastQC (https://www.bioinformatics.babraham.ac.uk/projects/fastqc/ (accessed on 5 July 2022)). To remove low-quality reads and the “contaminant” host reads from the metagenomic sequences, reads were trimmed using trimmomatic [26] and then aligned with the genome of C57BL/6J mice and filtered using Kneaddata [27]. Clean metagenomic sequence reads were analyzed using the Kraken2 taxonomic sequence classification approach on a standard Kraken database comprising all complete bacterial, viral, and archeal genomes in RefSeq [28]. The abundance of the various species was estimated using Bracken [29]. For functional classification, the reads were concatenated and then processed with default settings using HUMAnN3 [30]. Briefly, the reads were mapped to uniref90 database using DIAMOND [31], and the relative abundance of each gene family was quantified. The uniref-based gene family abundances were regrouped to KEGG orthologous groups, which were further used to make inferences on metabolic pathways. 

### 2.11. Statistics

All data values are expressed as mean ± standard deviation (SD) or median and interquartile range. Graphical illustrations and two-tailed unpaired *t*-test analyses were performed using GraphPad Prism 6.0 (GraphPad, San Diego, CA, USA).

## 3. Results

### 3.1. Early Life PCB 126 Exposure Persistently Affects AHR Signaling and Liver Nucleotide Metabolism in Both CTRL and HFD-Fed Mice in Adulthood

Early life PCB 126 exposure (24 μg/kg body weight per day for five days) had no persistent effect on body weight or liver histological analysis in mice fed a CTRL or HFD in adulthood (Appendix A). As expected, and confirming the considerably long half-life of PCB 126 in the rodents [32], increased mRNA expression of AHR target genes was observed in liver and ileal samples from CTRL and HFD-fed mice at four months after PCB 126 exposure (Figure 1B and Appendix A). ^1^H NMR analysis showed significantly higher levels of liver purines and pyrimidines from both CTRL and HFD-fed mice with PCB 126 exposure in adulthood (Figure 1C,D). Early life PCB 126 exposure resulted in the significantly higher levels of adenosine monophosphate (AMP), adenosine diphosphate (ADP), adenosine triphosphate (ATP), xanthine, and cytidine 5′-monophosphate (CMP) in the liver of mice fed with HFD and higher levels of xanthine and uridine in the liver of mice fed with CTRL in adulthood (Figure 1C,D). 

### 3.2. Early Life PCB 126 Exposure Causes Inflammatory Response in HFD-Fed Mice in Adulthood

No significant changes in blood biochemical markers were observed in CTRL or HFD-fed mice with early life PCB 126 exposure (Figure 2A). Early life PCB 126 exposure resulted in the significantly higher ratios of oxidized glutathione (GSSG) to reduced glutathione (GSH) in the liver of mice with HFD but not in CTRL-fed mice in adulthood (Figure 2B). Early life PCB 126 resulted in significant changes in two serum cytokines, including a lower level of leukemia inhibitory factor (LIF) and a higher level of lipopolysaccharide-induced CXC chemokine (LIX) in mice with HFD later in life but no changes in mice with CTRL (Figure 2C and Appendix A). The gene expression of the proinflammatory cytokines in adipose tissues, including tumor necrosis factor alpha (*Tnfα*) and NAD(P)H quinone dehydrogenase 1 (*Nqo1*), was significantly increased in early life PCB 126 exposure mice with HFD but not in CTRL-fed mice in adulthood (Figure 2D and Appendix A). 

### 3.3. Early Life PCB 126 Exposure Does Not Worsen the Effects of HFD Related to Adiposity in Adulthood

Oil red O staining of the liver showed no obvious changes, such as increased lipid accumulation and droplet size, in both CTRL and HFD-fed mice in adulthood with PCB 126 exposure (Figure 3A). Consistently, the quantitative ^1^H NMR analysis and triglyceride assay kit showed no significant changes in hepatic lipids, including fatty acids, in CTRL or HFD-fed mice in adulthood with early life PCB 126 exposure (Figure 3B–D). Consistent with the metabolomics data, no significant changes in the mRNA expression of genes involved in de novo fatty acid biosynthesis were observed in the liver from CTRL or HFD-fed mice with PCB 126 exposure (Figure 3E). 

### 3.4. Early Life PCB 126 Exposure Aggravates Impaired Glucose Homeostasis in HFD-Fed Mice in Adulthood

POPs have recently been linked with the development of type 2 diabetes and other metabolic diseases [33]. Quantitative ^1^H NMR analysis revealed that early life PCB 126 exposure resulted in significantly higher levels of metabolites involved in the tricarboxylic acid (TCA) cycle, including pyruvate, aspartate, fumarate, succinate, citric acid, glutamate, and α-ketoglutarate, in the liver from HFD-fed mice in adulthood (Figure 4A). Notably, the glucose tolerance test showed impaired glucose tolerance in HFD-fed mice later in life with early life PCB 126 exposure (Figure 4B). No significant changes were observed in metabolites involved in the TCA cycle or glucose tolerance test in CTRL-fed mice with early life PCB 126 exposure (Figure 4A,B). The mRNA expression of genes involved in gluconeogenesis was also significantly higher in the liver of HFD-fed mice with early life PCB 126 exposure (Figure 4C). Together, these results indicated that early life exposure to PCB 126 aggravates impaired glucose homeostasis in HFD-fed mice in adulthood.

### 3.5. Early Life PCB 126 Exposure Results in Diet-Dependent Changes in Bacteria Community, Gene Levels, and Metabolism in Adulthood

In order to investigate the influence of early life PCB 126 exposure on the gut microbiota with two diets, metagenomics and ^1^H NMR- and mass-spectrometry-based metabolomics analysis was performed. Metagenomic data showed that early life PCB 126 exposure resulted in significant changes in bacteria community and gene levels that were specific in two different diets in adulthood (Figure 5A–C and Appendix A). Early life PCB 126 exposure resulted in a significant decrease in the relative abundance of *Bacteroidetes* phyla and a higher ratio of *Firmicutes*/*Bacteroidetes* in cecal contents from mice with HFD in adulthood (Figure 5B and Appendix A). No pronounced effect in the ratio of *Firmicutes* to *Bacteroidetes* but significant changes in *Firmicutes* and *Verrucomicrobia* were observed in CTRL-fed mice with early life PCB 126 exposure (Figure 5B and Appendix A). Early life PCB 126 exposure resulted in significant decreases in the relative abundances of genera *Lacrimispora*, *Anaerocolumna*, *Anaerobutyricum*, *Anaerostipes*, *Mediterraneibacter*, *Blautia*, *Butyrivibrio*, *Bacillus*, *Clostridioides*, *Hungatella*, *Eubacterium*, *Lachnoclostridium*, *Ruthenibacterium*, *Roseburia*, *Streptococcus*, and *Caproiciproducens* but increases in the relative abundance of genus *Akkermansia* in CTRL-fed mice; and significant decreases in the relative abundances of genera *Muribaculum*, *Duncaniella*, *Bacteroides*, *Parabacteroides*, and *Prevotella* but increases in the relative abundances of genera *Romboutsia* and *Adlercreutzia* in HFD-fed mice (Figure 5A and Appendix A). We also analyzed the microbial metabolic pathway based on the KEGG database. Early life PCB 126 resulted in the induced expression of genes involved in L-histidine degradation, L-lysine biosynthesis, inosine-5′-phosphate biosynthesis, ADP-L-glycero-beta-D-manno-heptose biosynthesis, Calvin–Benson–Bassham cycle, TCA cycle, superpathway of pyridoxal 5’-phosphate biosynthesis and salvage, pyridoxal 5’-phosphate biosynthesis, tetrapyrrole biosynthesis, flavin biosynthesis, and phosphopantothenate biosynthesis but reduced expression of genes involved in stachyose degradation and purine ribonucleosides degradation in CTRL-fed mice (Figure 5C). Early life PCB 126 exposure reduced the expression of modules for the superpathway of L-aspartate and L-asparagine biosynthesis, L-histidine biosynthesis, L-arginine biosynthesis, phosphopantothenate biosynthesis, and the superpathway of coenzyme A biosynthesis in HFD-fed mice (Figure 5C). 

The influence of early life PCB 126 exposure on bacterial metabolites from mice fed with two diets was also investigated. We quantified the levels of common bacterial metabolites in the urine, feces, and cecal content samples using ^1^H NMR and UPLC-MS/MS analysis (Figure 5D–F and Appendix A). Early life PCB 126 exposure resulted in significantly lower levels of urine microbial metabolites, such as hippurate and phenylacetylglycine (PAG), in the mice with two diets in adulthood (Figure 5D). Early life PCB 126 resulted in a significantly higher level of cecal glycerol in HFD-fed mice later in life but not in CTRL-fed mice (Figure 5E). Early life PCB 126 did not exhibit a pronounced effect on cecal short-chain fatty acids (SCFAs) in both CTRL and HFD-fed mice, which was not consistent with what we observed in mice fed with standard chow diet [15] (Appendix A). The variation could be due to the different composition of gut microbiota and reduced formation of SCFAs with HFD and purified diet (CTRL) compared to the chow diet, which is rich in fermentable fiber [34]. The UPLC-MS/MS analysis of bile acid composition showed significantly increased unconjugated and conjugated bile acids in feces from both CTRL and HFD-fed mice in adulthood with PCB 126 exposure (Figure 5F and Appendix A). 

## 4. Discussion

Early life exposure to environmental pollutants is an increasing global problem, adversely and continuously affecting human health [14]. Over the last 50 years, the diet pattern of humans has shifted due to a rapid transition to saturated fat and refined sugars [35]. Accumulating evidence suggests that the connections between diet and environmental exposures are complex and important [19,20,21]; however, the interactions between diet and early life environmental exposure at doses not associated with overt toxicity have not been fully investigated. In this study, we systematically investigated the interaction between early life PCB 126 exposure and mice eating a CTRL or HFD. We determined diet-independent and diet-dependent metabolic and bacterial changes caused by early life exposure to PCB 126. We also demonstrated that early life PCB 126 exposure worsens HFD-induced impaired glucose homeostasis without worsening the effects of HFD related to adiposity in adulthood.

The diet-independent metabolic changes induced by early life PCB 126 exposure were observed in mice later in life. PCB 126, the most potent dioxin-like PCB, shows persistent AHR activation at four months after early life exposure in mice with CTRL and HFD due to its long half-life [32,36]. The significantly higher levels of liver nucleotides were observed with PCB 126 exposure in both CTRL and HFD-fed mice, which has also been reported with air pollution and PCB 126 exposure mice fed with standard chow diet [15,37]. To ensure the accuracy of DNA and RNA synthesis for replication, transcription, and translation, nucleotide metabolism is tightly regulated at all levels to maintain constant pools of pyrimidines and purines [38]. The disruption in nucleotide synthesis by PCB 126 exposure might support the unrestrained proliferation, regardless of diet status, which is usually observed in cancer cells and virus-infected cells [38]. 

Early life PCB 126 exposure worsens HFD-induced impaired glucose homeostasis without worsening the effects of HFD related to adiposity in adulthood. Emerging evidence suggests that POPs are involved in key mechanisms linking obesity and diabetes [39]. Our results show that early life PCB 126 worsens HFD-induced impaired glucose homeostasis in adulthood characterized by glucose intolerance and elevated in gluconeogenesis and the TCA cycle. A similar observation has been reported in a long-term PCB 126 exposure to adult mice that showed systemic effects associated with early end point of diabetes, including insulin resistance and impaired glucose metabolism induced by PCB 126 [10]. Future studies, including hepatic and systemic insulin sensitivity, are indeed warranted to fully capture the impact of changes in glucose homeostasis with early life PCB 126 exposure and HFD. It is interesting to note that PCB 126 and PCB 153 [19] both worsen diet-induced obesity/metabolic syndrome, suggesting the PCB effects on obesity/diabetes may depend more on diet interactions than on the congeners themselves. More work is needed to determine the role of ligand-activated transcription factors, such as the AHR, in relationship with environmental pollutants and the metabolic syndrome. Notably, the reduction in serum LIF and increase in serum LIX were observed in mice with HFD induced by early life PCB 126 exposure, suggesting an inflammatory response caused by early life PCB 126 that can contribute to the dysfunctional regulation of caloric intake and energy expenditure commonly present in obesity [40,41]. This notion is supported by the observation of up-regulated gene expression of several inflammatory mediators in adipose from HFD-fed mice with early life PCB 126 exposure. These are further accompanied by the significantly higher ratio of liver GSSG to GSH in HFD-fed mice with early life PCB 126 exposure, suggesting a marker of oxidative stress that is considered to be involved in various diseases, including obesity and diabetes [42]. Further evidence, such as H&E and immunohistochemical staining, is warranted to determine inflammation severity in the adipose tissue. Surprisingly, we did not observe increases in hepatic lipogenesis in mice fed a CTRL or HFD induced by early life PCB 126 exposure, which was reported in the experiments performed on human HepaRG liver cells, C57BL/6J mice fed with standard chow diet, and male Sprague Dawley rats fed with AIN-93G diet with PCB 126 exposure [15,43,44]. Emerging evidence demonstrated the direct role of AHR in liver lipid oxidation and the lipogenic pathway [45,46]. Our observation could be partially explained by the reduced availability of AHR ligands in the purified diets as compared to the standard chow diet [47] and the complexity of interactions between dietary nutrients and early life environmental pollutants. More studies are needed to rigorously test these variables using more variable animal models for the metabolic syndrome. 

Early life PCB 126 exposure resulted in diet-dependent changes in bacterial community structure and function in adulthood. It is increasingly clear that HFD alters the gut microbiota community structure, such as a decrease in *Bacteroidetes* and an increase in *Firmicutes,* which have been associated with obesity and chronic diseases [48,49]. We believe that diet not only plays a significant role in shaping the microbiome but also modifies the interactions between gut microbiota and the pollutants [50]. We observed a higher ratio of *Firmicutes* to *Bacteroidetes* in HFD-fed mice later in life with early life PCB 126 exposure, considering it a relevant marker of adverse changes in the gut microbiome community in obese subjects [49,51]. The decreased levels of genera *Blautia* and *Butyrivibrio*, which are butyrate-producing bacteria with anti-inflammatory properties [35,52], were observed in early life PCB 126 exposure mice with CTRL diet but not with HFD. The difference could be due to the significantly lower abundance of these bacteria with HFD [52]. Notably, early life PCB 126 exposure not only altered the bacteria community but also significantly changed microbial metabolism in adulthood. PCB 126 exposure resulted in reduced levels of urine hippurate in mice fed with CTRL and HFD, indicating decreased gut microbiome diversity, which is associated with multiple disorders, including the metabolic syndrome [53]. Microbial biosynthesis of amino acids in the human gastrointestinal tract plays a potential role in the host’s amino acid homeostasis [54]. The significant changes in microbial amino acid metabolism seen in both CTRL and HFD-fed mice with early life PCB 126 exposure suggest a disruption between microbial activity and host amino acid and energy homeostasis, which may be associated with the development of obesity [55]. Interestingly, a reduction in coenzyme A biosynthesis was also observed in HFD-fed mice with early life PCB 126 exposure, possibly indicating a disruption in the synthesis and degradation of fatty acids and phospholipids [56]. Consistently, we observed significantly higher levels of cecal glycerol with early life PCB 126 exposure in HFD-fed mice, suggesting the disturbance of fat absorption in the small intestine [57]. Similar observation has been reported with higher glycerol in fecal samples of patients suffering from Crohn’s disease, suggesting gut inflammation caused by the destruction of microbiota that is commonly associated with obesity [58]. More research is needed focusing on microbial fatty acid and lipid metabolism pathways to better understand the interaction with pollutant exposure and HFD. Furthermore, early life PCB 126 exposure resulted in elevated bile acid biosynthesis in mice later in life in both CTRL and HFD-fed mice, which could be due to the remodeling of gut microbiota and persistent AHR activation by PCB 126. Similar observations were seen in our previous studies showing significantly higher levels of bile acids in early life at five days of PCB 126 exposure in mice fed with standard chow diet [15].

## 5. Conclusions

Collectively, these data demonstrate that early life PCB 126 exposure persistently and significantly affects AHR signaling and liver nucleotide metabolism in mice fed with a CTRL or HFD in adulthood. Notably, we determined that early life PCB 126 worsened HFD-induced impaired glucose homeostasis characterized by glucose intolerance and elevated gluconeogenesis and TCA cycle flux, as well as changes in serum cytokines and adipose inflammatory genes, without worsening the effects of HFD related to adiposity in adulthood. Early life PCB 126 exposure resulted in diet-dependent changes in bacterial community structure and function later in life. These results provide a better understanding of the interactions between diet and early life environmental exposures and demonstrate that diet is a critical factor in the biochemical consequences of early life POP exposure involving the development of metabolic diseases.

## Figures and Tables

**Figure 1 metabolites-12-00894-f001:**
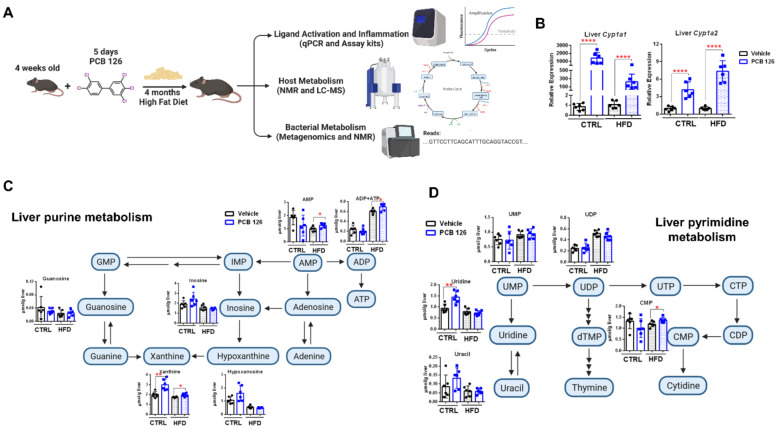
Effects of early life PCB 126 on AHR signaling and liver nucleotide metabolism in mice with CTRL or HFD. (**A**) Experimental schematic for determining the interaction between diet and early life PCB 126 exposure. (**B**) qPCR analysis of mRNA levels of AHR targeted genes in the liver from CTRL or HFD-fed mice with vehicle or PCB 126 exposure. (**C**,**D**) ^1^H NMR analysis of liver purines (**C**) and pyrimidines (**D**) from CTRL or HFD-fed mice with vehicle or PCB 126 exposure. Values are means ± S.D. (n = 6 per group). * *p* < 0.05, ** *p* < 0.01, **** *p* < 0.0001 compared to vehicle. AMP, adenosine monophosphate; ADP, adenosine diphosphate; ATP, adenosine triphosphate; CDP, cytidine 5′-diphosphate; CMP, cytidine 5′-monophosphate; CTP, cytidine 5′-triphosphate; GMP, guanosine monophosphate; IMP, inosine monophosphate; UMP, uridine monophosphate; UDP, uridine 5′-diphosphate; UTP, uridine 5′-triphosphate.

**Figure 2 metabolites-12-00894-f002:**
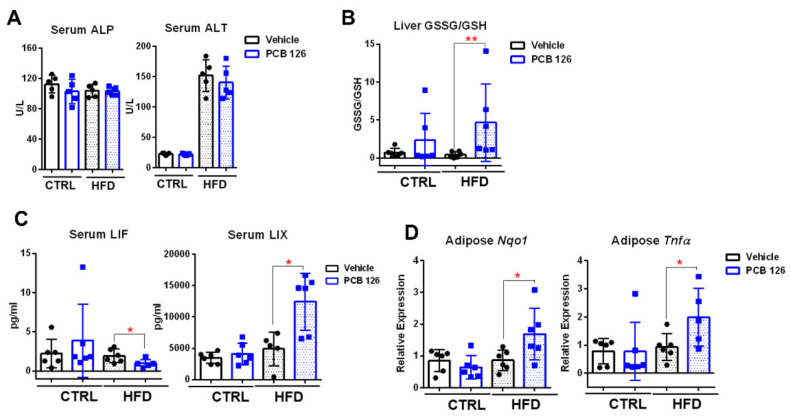
Effects of early life PCB 126 on inflammatory indicators in mice with CTRL or HFD. (**A**) Serum concentrations of alanine transaminase (ALT) and alkaline phosphatase (ALP) from CTRL or HFD-fed mice with vehicle or PCB 126 exposure. (**B**) Ratio of oxidized glutathione (GSSG) to reduced glutathione (GSH) in the liver from CTRL or HFD-fed mice with vehicle or PCB 126 exposure. (**C**) Serum cytokines, including leukemia inhibitory factor (LIF) and lipopolysaccharide-induced CXC chemokine (LIX), from CTRL or HFD-fed mice with vehicle or PCB 126 exposure. (**D**) qPCR analysis of mRNA levels of inflammatory cytokines in the adipose from CTRL or HFD-fed mice with vehicle or PCB 126 exposure. Values are means ± S.D. (n = 6 per group). * *p* < 0.05, ** *p* < 0.01 compared to vehicle.

**Figure 3 metabolites-12-00894-f003:**
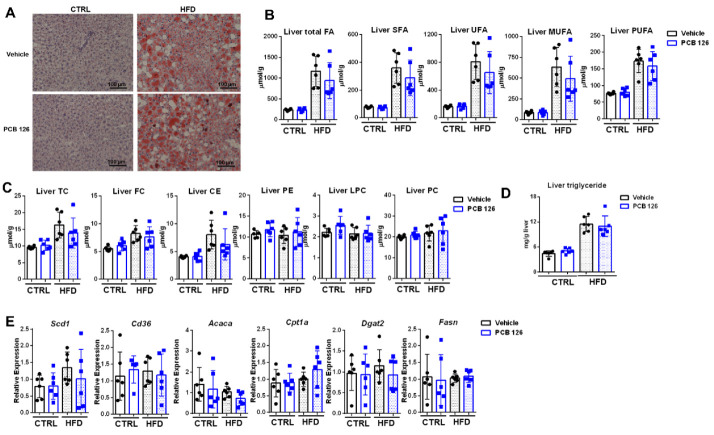
Effects of early life PCB 126 on liver lipid metabolism in mice with CTRL or HFD. (**A**) Oil Red O staining in the liver of CTRL or HFD-fed mice with vehicle or PCB 126 exposure. (**B**,**C**) Quantitative ^1^H NMR analysis of liver fatty acid (**B**) and lipid (**C**) profiling from CTRL or HFD-fed mice with vehicle or PCB 126 exposure. (**D**) Liver triglyceride levels from CTRL or HFD-fed mice with vehicle or PCB 126 exposure. (**E**) qPCR analysis of mRNA encoding de novo fatty acid biosynthesis in liver from CTRL or HFD-fed mice with vehicle or PCB 126 exposure. Values are means ± S.D. (n = 6 per group). TC, total cholesterol; FC, free cholesterol; CE, cholesterol ester; PE, phosphatidylethanolamine; LPC, lysophosphatidylcholine; PC, phosphatidylcholine; FA, fatty acid; SFA, saturated fatty acid; UFA, unsaturated fatty acid; MUFA, monosaturated fatty acid; PUFA, polyunsaturated fatty acid.

**Figure 4 metabolites-12-00894-f004:**
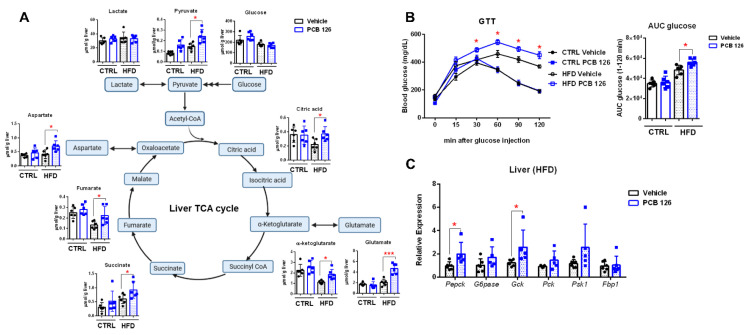
Effects of early life PCB 126 on glucose metabolism in mice with CTRL or HFD. (**A**) Quantitative ^1^H NMR analysis of metabolites involved in tricarboxylic acid (TCA) cycle flux in liver from CTRL or HFD-fed mice with vehicle or PCB 126 exposure. (**B**) Blood glucose levels following injection of glucose and the area under the curve (AUC) from CTRL or HFD-fed mice with vehicle or PCB 126 exposure. (**C**) qPCR analysis of mRNA encoding gluconeogenesis in liver from HFD-fed mice with vehicle or PCB 126 exposure. Values are means ± S.D. (n = 6 per group). * *p* < 0.05, *** *p* < 0.001 compared to vehicle.

**Figure 5 metabolites-12-00894-f005:**
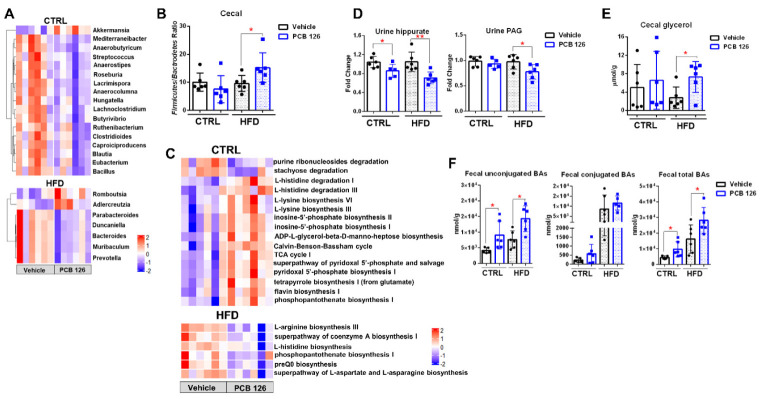
Effects of early life PCB 126 on bacteria community and metabolism in mice with CTRL or HFD. (**A**) Heatmap representation of the abundance of cecal bacteria from genera that were significantly changed from CTRL or HFD-fed mice with vehicle or PCB 126 exposure. (**B**) Ratio of *Firmicutes* to *Bacteroidetes* in cecal contents from CTRL or HFD-fed mice with vehicle or PCB 126 exposure. (**C**) Analysis of KEGG pathway abundance that significantly changed in the cecal microbiota from CTRL or HFD-fed mice with vehicle or PCB 126 exposure. (**D**) ^1^H NMR analysis of urine hippurate and phenylacetylglycine (PAG) from CTRL or HFD-fed mice with vehicle or PCB 126 exposure. (**E**) ^1^H NMR analysis of cecal glycerol from CTRL or HFD-fed mice with vehicle or PCB 126 exposure. (**F**) Quantitative UPLC-MS analysis of bile acids in the feces from CTRL or HFD-fed mice with vehicle or PCB 126 exposure. Values are means ± S.D. (n = 6 per group). * *p* < 0.05, ** *p* < 0.01 compared to vehicle.

## Data Availability

The data presented in this study are available in article and Appendix A.

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
