# Peer review of "Early Life Polychlorinated Biphenyl 126 Exposure Disrupts Gut Microbiota and Metabolic Homeostasis in Mice Fed with High-Fat Diet in Adulthood"

_metabolites, 2022, doi:10.3390/metabo12100894_

Round 1
Reviewer 1 Report
The manuscript “Early life polychlorinated biphenyl 126 exposure disrupts gut microbiota and metabolic homeostasis in mice fed with high fat diet in adulthood” describes disruptions of gut microbiome and metabolic homeostasis induced by exposure of PCB 126. Although this study was designed well and performed precisely, there are a few points that need to be addressed prior to publication.
Point 1: In the introduction, the rationale of the study design should be described more. For example, why the PCB 126 was exposed in early life, not in adulthood? Why the high fed diet was performed?
Point 2 In the results, a few metabolome data were not shown reproducibly with the previous study (reference no. 14 in the manuscript). For example, SCFAs were changed between PCB 126 exposed group and control group in previous study, which were not changed in that study.
Point 3: In Figure 5A, the name of phylum should be denoted, because readers would know that which genus is classified into Firmicutes or Bacteriodetes.
Point 4: In Figure 5 and L287-298, urine hippurate and fecal bile acids, which are microbial metabolites, seemed to be diet-independent changes, which means microbial changes were not associated with the mechanism of impaired glucose homeostasis.
Point 5: Why didn’t the authors measure the endogenous CYP1A1/2 metabolites, for example, kynurenine? It is well known that these metabolites are related to gut microbiome and lipid metabolism.
Author Response
Point 1: In the introduction, the rationale of the study design should be described more. For example, why the PCB 126 was exposed in early life, not in adulthood? Why the high fed diet was performed?
Response: We addressed these points in the introduction.
Please see the 2nd paragraph on Page 3: “However, the long-term consequences of PCB 126 exposure at vulnerable life stages such as early childhood has received less attention [14]. Recently, we demonstrated that early life PCB 126 exposure resulted in metabolic abnormalities and microbiota changes in mice fed with standard chow diet in adulthood, providing evidence for an association between early life environmental pollutant exposure and increased risk of metabolic disorders later in life [15]. ”
Please see the 1st paragraph on Page 4: “The increased risk of diabetes/obesity is positively correlated to PCBs exposure [16-18]. Studies have reported that 2,2′,4,4′,5,5′-hexachlorobiphenyl (PCB 153), a non-dioxin like PCB, is a diet-dependent obesogen, which may worsen non-alcoholic fatty liver disease via adipokine dysregulation and altered hepatic lipid metabolism in mice [19]. A recent study reported that PCB 126 exposure alters the global epi-transcriptome in a mouse model of nonalcoholic steatohepatitis [20]. Recent studies also indicated that two doses of PCB 126 exposure (326 μg/kg body weight) dramatically disrupted the gut microbiota and host metabolism and increased intestinal and systemic inflammation in a low density lipoprotein receptor knockout (Ldlr-/-) mouse model fed a high cholesterol diet [21]. Likewise, diet-toxicant interactions appear to be important in diet-induced obesity/diabetes, but remain largely unexplored, particularly for early life environmental exposure.”
Point 2: In the results, a few metabolome data were not shown reproducibly with the previous study (reference no. 14 in the manuscript). For example, SCFAs were changed between PCB 126 exposed group and control group in previous study, which were not changed in that study.
Response: In reference no. 14, PCB 126 exposure resulted in lower levels of SCFAs from mice fed with standard chow diet. In this study, we did not observe significant changes in SCFAs from mice fed with high fat diet or purified diet by PCB 126 exposure. We added discussion about this variation. Please see the 2nd paragraph on Page 12 “ Early life PCB 126 did not exhibit a pronounced effect on cecal short chain fatty acids (SCFAs) in both CTRL and HFD-fed mice, which were not consistent with what we observed in mice fed with standard chow diet [15] (Supplementary Figure S5B). The variation could be due to the different composition of gut microbiota and reduced formation of SCFAs with HFD and purified diet (CTRL) compared to chow diet that is rich in fermentable fiber [34]. ”
Point 3: In Figure 5A, the name of phylum should be denoted, because readers would know that which genus is classified into Firmicutes or Bacteriodetes.
Response: We added this information in Supplementary Tables S4-5.
Point 4: In Figure 5 and L287-298, urine hippurate and fecal bile acids, which are microbial metabolites, seemed to be diet-independent changes, which means microbial changes were not associated with the mechanism of impaired glucose homeostasis.
Response: We observed diet-independent and diet-dependent bacterial changes caused by PCB 126 exposure. The changes in urine hippurate and fecal bile acids were observed in both CTRL and HFD, indicating decreased gut microbiome diversity and persistent AHR activation that are independent of diet status. However, we also observed diet-dependent changes in microbial metabolites. For example, significantly higher levels of cecal glycerol with PCB 126 exposure were observed only in HFD-fed mice, suggesting disturbance of fat absorption in the small intestine. Reduction in coenzyme A biosynthesis was observed only in HFD-fed mice with PCB 126 exposure, indicating a disruption in the synthesis and degradation of fatty acids and phospholipids (Please see Pages 16-17).
Point 5: Why didn’t the authors measure the endogenous CYP1A1/2 metabolites, for example, kynurenine? It is well known that these metabolites are related to gut microbiome and lipid metabolism.
Response: We did not measure tryptophan metabolites in this study. Tryptophan metabolites are endogenous AHR ligands, not endogenous CYP1A1/2 metabolites (Drug Metab Dispos 2015, 43: 1522-1535).
Reviewer 2 Report
The authors of the manuscript investigated the effects of early exposure of PCB 126 alone or combined with high-fat diet to C57BL/6J mice on AHR signaling, liver nucleotide metabolism, blood inflammatory markers, liver adiposity, glucose homeostasis, gut microbiome and gut microbial metabolism in adulthood of the mice. In general, this is an interesting study. The design of the experiments is appropriate; the data is sound; and the conclusion is supported by the results. No major concerns were identified; some minor issues need to be addressed.
1. The dosage, duration and time points of the treatment need to be justified.
2. The general metabolism of PCB 126 in the body should be described.
3. The implication/significance of the combined effects of PCB 126 and high-fat diet on tested measures should be discussed.
Author Response
1. The dosage, duration and time points of the treatment need to be justified.
Response: We added an experiment flow chart as Supplementary Figure S1.
- The general metabolism of PCB 126 in the body should be described.
Response: Thank you for the suggestion. PCB 126 is not metabolized in the body (Chemosphere 1989, 19: 809-812). We added more information in the Introduction Section of the manuscript as “Unchanged PCB 126 may remain in the body and stored for years mainly in the fat and liver [8].”
- The implication/significance of the combined effects of PCB 126 and high-fat diet on tested measures should be discussed.
Response: We added more discussion about the significance of studying diet-toxicant interactions in the Discussion section. Please see the 1st paragraph in Page 14: “Early-life exposure to environmental pollutants is an increasing global problem, adversely and continuously affecting human health [14]. Over the last 50 years, the diet pattern of humans has shifted due to a rapid transition to saturated fat and refined sugars [35]. Accumulating evidence suggests that connections between diet and environmental exposures are complex and important [19-21]; however, the interactions between diet and early life environmental exposure at doses not associated with overt toxicity have not been fully investigated.”
Reviewer 3 Report
The results support that early life PCB 126 exposure exacerbates glucose intolerance induced by HFD feeding, along with gut microbiota profile changes in mice. However, changes in critical organs that play important roles in glucose homeostasis were not clearly presented. Please specify if liver samples were collected under a fasting condition. Please consider performing experiments that examine alterations in hepatic and systemic insulin sensitivity, such as insulin tolerance test and the activation of insulin signaling pathway after insulin infusion. Also, please consider providing further evidence that shows changes in the inflammation severity in white adipose tissue, such as H&E and immunohistochemical staining.
Author Response
Response: Mice were not fasted before sacrifice. This information was added in the Methods section. We did not perform insulin tolerance tests in this study. We did perform glucose tolerance tests and measured metabolites from liver TCA cycle and liver mRNA expression of genes involved in gluconeogenesis to study glucose homeostasis. We did not do H&E or immunohistochemical staining on adipose tissue in this study. We did liver H&E staining and measured liver glutathione, blood biochemistry and cytokines, and expression of adipose proinflammatory cytokine mRNA to assess the overt toxicity and adipose inflammation. We added the limitations of current study and the direction for future studies. Please see the Page 15: “Future studies including hepatic and systemic insulin sensitivity are indeed warranted to fully capture the impact of changes in glucose homeostasis with early life PCB 126 exposure and HFD.” “Further evidence such as H&E and immunohistochemical staining is warranted to determine inflammation severity in the adipose tissue.”